# Can Drones Help Smallholder Farmers Improve Agriculture Efficiencies and Reduce Food Insecurity in Sub-Saharan Africa? Local Perceptions from Malawi

Christopher McCarthy [1,2,*], Yamikani Nyoni [2], Daud Jones Kachamba [2], Lumbani Benedicto Banda [3], Boyson Moyo [3], Cornelius Chisambi [4], James Banfill [5] and Buho Hoshino [6,*]

1. Zanvyl Krieger School of Arts & Sciences, Johns Hopkins University, Baltimore, MD 21218, USA
2. Department of Forestry, Bunda College of Agriculture Campus, Lilongwe University of Agriculture and Natural Resources (LUANAR), Lilongwe P.O. Box 219, Malawi; nyoni@climatesmartlabs.org (Y.N.); dkachamba@luanar.ac.mw (D.J.K.)
3. Department of Environment and Natural Resources, Bunda College of Agriculture Campus, Lilongwe University of Agriculture and Natural Resources (LUANAR), Lilongwe P.O. Box 219, Malawi; lbbanda@bunda.luanar.mw (L.B.B.); bmoyo@luanar.ac.mw (B.M.)
4. Graduate School of Global Environmental Studies, Kyoto University, Kyoto 606-8501, Japan; cornelius.chisambi.42s@st.kyoto-u.ac.jp
5. Institute of Asian Studies, Chulalongkorn University, Bangkok 10330, Thailand; james.banfill@alumni.psu.edu
6. Lab of Environmental Remote Sensing, Department of Environmental Sciences, College of Agriculture, Food and Environment Sciences, Rakuno Gakuen University, Ebetsu 069-8501, Japan
* Correspondence: cmccar27@jh.edu (C.M.); aosier@rakuno.ac.jp (B.H.)

**Abstract:** Smallholder farmers in sub-Saharan Africa play a vital role in achieving food security and nutrition, yet they are often overlooked by development policies and lack access to the technology and information needed to increase their agricultural productivity. This is particularly true in Malawi, where smallholder farmers make up over 80% of the population and face a range of risks and challenges, including vulnerability to climate change, that threaten their livelihoods, food security, and nutrition. While drones and precision agriculture technology have had a significant impact on agriculture in high-income countries, their application by smallholder farmers in low-income countries is not well understood. This study, conducted in 2022, examines how drones can assist smallholder farmers in increasing their agricultural productivity and food security in Malawi. It explores how smallholders perceive the use of drones and the potential benefits and limitations of using drones in their farming operations. A unique aspect of this study aims to understand smallholders' perceptions of open data and data privacy. The results show that when smallholder farmers interact with drone data, they have a better understanding of their farm and are able to make more informed decisions that use fewer inputs and reduce production costs. Overall, this study demonstrates the potential for drones to assist smallholder farmers improve their on-farm knowledge, increase agricultural productivity, and mitigate the risks and challenges they face, leading to improved livelihoods and a more sustainable and secure food supply. Policymakers can promote the adoption of drone technology among smallholder farmers by developing policies that encourage public–private partnerships to create affordable, locally adapted drone technologies and programs that meet their unique needs, while also ensuring responsible use of drones in agriculture through regulations that address concerns about data privacy and security.

**Keywords:** unmanned aerial vehicles; open data; agriculture extension; smallholder agriculture; sustainable agriculture; climate-smart agriculture; sustainable development goals

## 1. Introduction

Smallholder farms in sub-Saharan Africa provide the majority of the region's food supply and are crucial for sustaining livelihoods, ensuring food security, and improving

nutrition [1–3]. Despite this, agricultural productivity has remained stagnant compared to significant increases in other parts of the world [4]. Since 1960, the total food production in sub-Saharan Africa has remained constant, while it has increased by 50% globally [5]. Global food demand is expected to rise by 60% by 2050, with an even greater increase projected in sub-Saharan Africa [6]. Currently, crops in sub-Saharan Africa yield only 20% of their potential, and the average farm performs at only around 40% of its capacity [7,8]. If current trends continue, many sub-Saharan countries may only produce 13% of their food needs by 2050, potentially leading to increased food insecurity and malnutrition [9,10]. Globally, farming practices, particularly in arid regions, are increasingly strained by the impacts of climate change and anthropogenic activities [11]. These trends, mainly attributed to land fragmentation, climate change, and a rapidly growing population, can further threaten the productivity and sustainability of smallholder farmers [12]. Improving the productivity of smallholder farms in sub-Saharan Africa is a significant opportunity to make progress toward achieving the Sustainable Development Goals (SDGs) of ending poverty and hunger, specifically SDG 1 (no poverty) and SDG 2 (zero hunger) [13–15].

The various challenges facing smallholder farmers in sub-Saharan Africa highlight the urgent need for innovation in the agricultural sector. Historically, improved agricultural productivity has been achieved through better economic incentives for farmers, investments in agricultural research, and the widespread adoption of new technologies, including increasingly digital tools and data [4,16]. In developed high-income countries, the digitization of agriculture, including precision farming, has transformed the way food is grown and land is managed, leading to increased productivity and crop yields, lower input costs, and reduced environmental impacts of on-farm activities. As the cost of accessing digital farming technology has decreased, tools, such as mobile phone apps; field sensors; and remote sensing technologies, such as drones, have become more affordable and accessible, offering an opportunity for farmers in sub-Saharan Africa to bypass the digital divide and transform the struggling agriculture sector, thereby improving the livelihoods of smallholder farmers [17,18]. While the potential benefits of digital agriculture and precision farming are significant, the uptake of these technologies by smallholder farmers in sub-Saharan Africa is low and not well understood.

In recent years, the potential for drones to revolutionize agriculture has received significant attention [19–21]. Drones can assist farmers in tasks such as field mapping, crop monitoring, evaluation, and classification [22]. Drone images can be used for detecting plant stress [23,24], managing weeds [25], counting crops [26], and estimating soil nutrients [27,28]. Drone technology has the potential not only to boost agricultural productivity but also to support climate-smart agriculture by facilitating better resource management and adaptation to climate change [29,30]. As technology advances and sensors and batteries improve, the capabilities of drones in agriculture expand, making them an increasingly valuable tool for farmers [31]. The decreasing cost and increasing efficiency of drones make them an appealing option for those in the agricultural industry. While the use of drones in agriculture is growing globally, adoption of this technology in Africa remains low [32]. Factors contributing to this low uptake can include cost constraints, a lack of trained personnel, and insufficient infrastructure [33]. Some countries, such as Mozambique, have demonstrated that extension workers using drones can help farmers make informed decisions to improve crop-water efficiency and yields [32]. Despite this, research on the use of drones in agriculture, including incorporating drone images as data sources, is scarce in sub-Saharan Africa, and more research is needed to better understand smallholder farmers' perceptions on use of drones [34,35]. Previous studies have primarily focused on the adoption of drones in agriculture in developed countries, leaving a knowledge gap in the context of smallholder farmers in sub-Saharan Africa.

Malawi, like many countries in sub-Saharan Africa, has a low adoption rate of drones in the agricultural sector. The low utilization of drones for agricultural purposes may be due to a lack of knowledge about the technology among smallholder farmers and the high cost of the technology; however, drones present an opportunity for the country to improve

its agricultural sector through enhanced delivery of extension services and improved land management. As many smallholder farmers in Malawi live in rural areas with limited access to technology transfer, they have had few opportunities to use drones in agriculture.

To better understand the challenges and opportunities for drones to improve on-farm efficiencies of smallholders in sub-Saharan Africa, this study explores how farmers in Malawi perceive and interact with the use of drones and data from drone imagery. An additional focus of this study aims to explore farmers' perceptions of open data and data privacy. Smallholder farmers have a lot to benefit from the decentralized sharing of data, including the ability to better understand their farms and make more informed decisions about their operations; however, there are concerns about the potential for data misuse and the impact on privacy, which can further exacerbate the vulnerabilities of smallholder farmers in the context of the complex financial, commodity, and information flows within agri-food chains [36]. To ensure the responsible use of data in agriculture and protect the interests of smallholder farmers, it is important to address these privacy and security concerns and establish clear guidelines and regulations for data sharing. By surveying farmers about their views on open data and data privacy, this study aims to better understand the potential benefits and risks of using open data in agriculture and to provide recommendations for its responsible and ethical use. By highlighting the direct relationship between new technologies, such as drones, and their impact on reducing food insecurity and boosting agricultural productivity, this study emphasizes the importance of technology adoption for the future of smallholder farming in sub-Saharan Africa.

The specific objectives of the study were to (a) analyze farming practices using drones, (b) introduce farmers to drone-based farm management, (c) survey farmers' perceptions on the use of drones in farm management and agriculture production, and (d) explore farmers' perceptions of open data and data privacy.

The study sought to answer the following questions:

1.  What can the data from drones reveal about smallholder farming practices?
2.  How do farmers perceive the use of drones and data to assist with on-farm decision-making?
3.  How can farmers use drones to complement their existing knowledge to increase productivity and reduce production costs?
4.  How do smallholder farmers perceive open data and data privacy?

## 2. Materials and Methods

Data were collected through a combination of methods, including focus group discussions, individual interviews, farm visits and drone imagery collection. The focus group interviews were conducted with farmers to gather their perspectives on drones and their potential use in agriculture. Farm surveys were conducted at each farm to assess the layout and planning of the farm and to understand how farmers were utilizing their land for maximum productivity. During the farm surveys, a drone was flown to map the farm. The results of the drone analysis were shared with the farmers, who were asked to compare their understanding of their farms before and after seeing the overview provided by the drone data, which included orthomosaic images, crop health profiles, and elevation information. The farmers were asked if the use of orthomosaic images and plant health profiles helped to fill any gaps they had identified in their initial farm assessments. At the end of the study, all of the collected data on the farms were shared with the farmers to aid in management planning and future follow-up. Finally, farmers were asked about their perceptions of data sharing and privacy.

Interviews and farm surveys were conducted between September and October 2022 to gather data during the crop season when farmers were actively engaged in their farming activities. This timing allowed for a more accurate assessment of the current state of their farms and management practices. Additionally, the timing of the survey provided the opportunity to assess the potential benefits of using drones for farm management and productivity during the crop season. The data collected during this time also allowed for

immediate feedback and analysis, which could help farmers to make timely decisions to improve their farming practices. The results of the survey could be used to inform future land management decisions and planning, particularly as they relate to the use of drones in agriculture.

### 2.1. Study Area

This study was conducted in Neno district, traditional authority Symon, southern Malawi (Figure 1). Located at 16°10′ S latitude and 35°10′ E longitude, Neno district borders Mozambique to the west, Ntcheu district to the north, Balaka and Zomba districts to the northeast, Blantyre district to the east, Chikwawa to the south, and Mwanza district to the southwest. The area is known for its subsistence and commercial farming, which supplies the markets along the M1 road (the main road connecting all regions in Malawi). The Shire River, which demarcates the boundary between Neno and Blantyre districts, supports irrigation farming, allowing farmers to produce crops throughout the year. The district has an average annual rainfall of 530 mm from November to March, with the majority of rain falling in December, January, and February. The annual average temperature is 24 °C. The Neno district is sparsely populated, with a population of approximately 161,985 people and a total area of 1469 square kilometers at an elevation of 672 m. Neno was targeted given its large number of smallholders and diverse farming approaches.

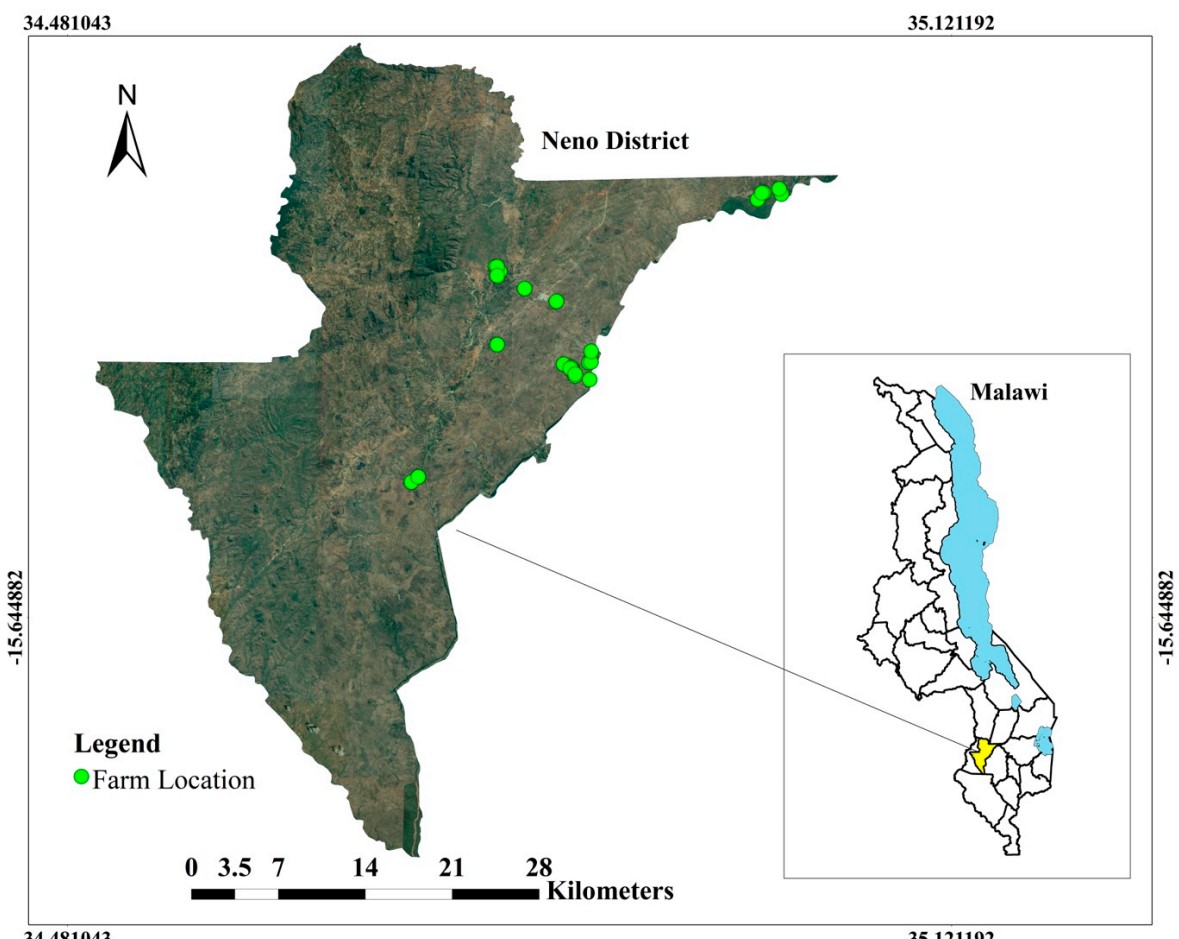

**Figure 1.** Map of study area.

### 2.2. Participant Selection

Participants for this study were selected through a combination of random sampling and recommendations from the local agricultural extension agent within the Neno district extension planning area. Some participants were chosen based on previous relationships

and their proximity to the study area, while others were identified through referrals from other farmers. In addition, we aimed to select participants from a variety of farming systems (e.g., irrigation, rainfed, etc.) and crop/management stages. In total, 30 smallholder farmers from 30 different farms participated in the study. We define a smallholder farmer as an individual or household who owns or manages a small area of land, typically less than five hectares, and relies predominantly on family labor to produce crops and/or raise livestock for subsistence and/or market purposes. Among the participants, 22 were male and 8 were female, with a distinction made between commercial and subsistence farmers. Commercial farming tends to be larger scale, more specialized, and heavily reliant on technology and external inputs, while subsistence farming is often smaller scale, diversified, and more reliant on traditional knowledge and local resources.

Overall, while the study's sampling strategy aimed to ensure a diverse range of participants, it is important to acknowledge the possible sampling error due to the limited sample size (30 smallholder farmers) and the study's focus on a single district within Malawi. As a result, the generalizability of the findings to other regions or districts within Malawi or other sub-Saharan African countries may be limited. Additionally, participants were selected through a combination of random sampling and recommendations from the local agricultural extension agent, which may introduce bias into the selection process. Therefore, the extent to which the study's findings can be generalized to other smallholder farmers in Malawi or other sub-Saharan African countries should be interpreted with caution. Nonetheless, the study's results provide valuable insights into the attitudes and perceptions of smallholder farmers towards drone technology in agriculture, particularly for different types of farmers. Table 1 provides an overview of the survey participants.

**Table 1.** Demographic characteristics of project participants.

|  | Characteristics | Commercial | Subsistence | Total |
|---|---|---|---|---|
| Role of the respondent | Owner | 4 (13%) | 19 (63%) | 23 (76%) |
|  | Manager/Laborer | 5 (17%) | 2 (7%) | 7 (24%) |
| Irrigation/Rainfed | Irrigation | 14 (47%) | 0 (0%) | 14 (47%) |
|  | Rainfed | 0 (0%) | 16 (53%) | 16 (53%) |
| Age | 18–24 | 1 (3%) | 0 (0%) | 1 (3%) |
|  | 25–34 | 0 (0%) | 16 (53%) | 16 (53%) |
|  | 35–44 | 5 (17%) | 2 (7%) | 7 (24%) |
|  | 45–55 | 1 (3%) | 3 (10%) | 4 (13%) |
|  | 55+ | 2 (7%) | 0 (0%) | 2 (7%) |
| Gender | Males | 6 (20%) | 16 (53%) | 22 (73%) |
|  | Females | 3 (10%) | 5 (17%) | 8 (27%) |
| Education | Primary | 2 (7%) | 12 (40%) | 14 (47%) |
|  | Secondary | 5 (17%) | 7 (23%) | 12 (40%) |
|  | Tertiary | 3 (10%) | 1 (3%) | 4 (13%) |
| Farm size (hectares) | <3 | 5 (17%) | 13 (43%) | 18 (60%) |
|  | 4–10 | 2 (7%) | 8 (27%) | 10 (34%) |
|  | 11–15 | 0 (0%) | 0 (0%) | 0 (0%) |
|  | 16–20 | 1 (3%) | 0 (0%) | 1 (3%) |
|  | 21–30 | 1 (3%) | 0 (0%) | 1 (3%) |

**Table 1.** *Cont.*

|  | Characteristics | Commercial | Subsistence | Total |
|---|---|---|---|---|
| Main crop | Maize | 3 (10%) | 21 (70%) | 24 (80%) |
|  | Tomatoes | 3 (10%) | 0 (%) | 3 (10%) |
|  | Bananas | 1 (3%) | 0 (%) | 1 (3%) |
|  | Eggplants | 2 (7%) | 0 (%) | 2 (7%) |
| Own/Rent land | Own | 7 (23%) | 16 (53%) | 23 (76%) |
|  | Rent | 2 (7%) | 5 (17%) | 7 (24%) |

*2.3. Interviews and Drone-Based Farm Surveys*

To understand the farmers' perceptions of drones and their potential use in agriculture, in-person interviews and farm surveys were conducted, in which aerial farm imagery was collected through drone flights and data shared with farmers.

2.3.1. Interviews with Farmers

To better understand how farmers perceive the use of drones in their farming practices, surveys and interviews were conducted with individual farmers covering various topics, such as the usefulness of orthomosaic images, plant health data, and elevation profiles, as well as concerns related to data sharing and privacy. Understanding the knowledge, perceptions, and attitudes of the potential adopter has been shown to play a key role in the uptake of agricultural technologies [37]. Surveys and interviews were conducted in both Chichewa and English, with particular care taken to use simple language and visuals to ensure understanding among all participants. The responses were systematically coded and analyzed to reveal common themes and patterns in the qualitative data.

2.3.2. Drone-Based Farm Surveys

A DJI Mavic 2 Pro drone equipped with a high-resolution camera (20 MP with 1-inch CMOS) was employed for all flights, while flight planning and mapping were conducted using the DroneDeploy software (www.dronedeploy.com (accessed on 6 May 2023)). The drone captured images of the farms from an elevation of 60 m, with 70% front overlap and 65% side overlap at a drone speed of 6 m/s. The orientation of the image blocks was achieved using the DroneDeploy software without the use of ground control points. While using ground control points (GCPs) can further enhance the accuracy of the image processing and the resulting map, GCPs were not used due to the limited time and resources available. Nonetheless, it was ensured that the drone flights were conducted under consistent weather conditions, and the drone was flown at a constant altitude, speed, and overlap rate. Additionally, the DroneDeploy software has been shown to produce accurate and reliable results in similar studies [38].

The farmers were actively involved in the drone survey process. Before the flight, the research team worked closely with the farm owners to accurately identify the boundaries of the farms to be surveyed. To save time and collect sufficient data, multiple neighboring farms were surveyed during each flight. Following the drone flights, the images were provided to the farmers, who were then asked to annotate the images with land use and identify on-farm issues, such as soil erosion, pests and diseases, and water stress. Farmers then discussed with agricultural extension agents about potential solutions for mitigating the issues identified from the drone imagery. The research team assisted the farmers by explaining the terms, effects, and usage in the local language to ensure accurate understanding and active participation in the discussion.

*2.4. Data Analysis*

2.4.1. Drone Imagery Analysis

After the drone flight, collected images were uploaded onto the DroneDeploy platform to produce an orthomosaic map. Farm profiles were created, which included plant health and elevation profiles, using tools provided on the platform (Mobile version-4.120.0 and Web Version-2.201.0). Plant health status was created based on the Visible Atmospheric Resistant Index (VARI) algorithm, which is commonly used for RGB cameras and remote estimation of vegetation performance [35]. Image processing took between 30 min to 2 h, depending on the size of the farm. After each field visit, data was processed the same day, ready to be discussed with the farmers on the next visit. This approach is consistent with the recommendations for real-time data analysis and feedback in precision agriculture [39].

2.4.2. Survey Result Analysis

The survey responses from farmers were analyzed using both descriptive and thematic analysis to identify common themes and patterns in the qualitative responses. Descriptive statistics were used to summarize the responses to each question, including mean, standard deviation, and frequency distributions. The thematic analysis involved a systematic process of coding and categorizing the responses to identify key themes, which were then further analyzed to provide insights into farmers' attitudes, perceptions, and behaviors towards drone technology. Overall, the combination of descriptive and thematic analysis provided a comprehensive understanding of the farmers' perspectives on drone technology and the factors that influence its adoption.

2.4.3. Data Sharing and Privacy

In order to analyze the survey responses related to data sharing and privacy, a combination of quantitative and qualitative methods were employed. The quantitative analysis involved summarizing the responses to closed-ended questions using descriptive statistics, such as frequency distributions and percentages. The open-ended questions were analyzed qualitatively by identifying common themes and patterns in the responses through content analysis.

## 3. Results

In this section, we present the findings of our study on the adoption of drone technology among smallholder farmers, which are organized into four main subsections. Section 3.1 gives an overview of the respondents' demographic characteristics, setting the context for their agricultural practices and backgrounds. In Section 3.2, we explore the outcomes of the farm surveys, demonstrating the practical application of drone technology in identifying on-farm issues and potential solutions. Section 3.3 examines the farmers' perceptions and attitudes towards the use of drone technology in agriculture, uncovering their level of interest and concerns about the technology. Lastly, in Section 3.4, we investigate the farmers' views on data sharing and privacy, emphasizing their willingness to share drone data and their concerns about the potential risks and benefits involved.

*3.1. Demographic Characteristics of the Respondents*

The majority of study participants were male, making up 73% of the total respondents (Table 1). Around 70% of the respondents were subsistence farmers, while the remaining 30% were commercial farmers. Sixteen farmers (53%) rely on rainfed agriculture, and fourteen (47%) also practiced irrigation farming. The farm sizes varied among the respondents, however, most farmers (60%) cultivated on farms that were less than 3 hectares in size. The results also indicated that most farmers (76%) owned the land they cultivated rather than renting it. In terms of education, the majority of participants had completed at least a primary education (47%). Forty percent completed secondary education and 13% of participants had completed tertiary education.

*3.2. Farm Surveys*

Examples of the drone images provided to the farmers and annotated images are provided below in Figure 2A–E. Distribution of on-farm issues identified in the drone images by the farmers is presented in Table 2.

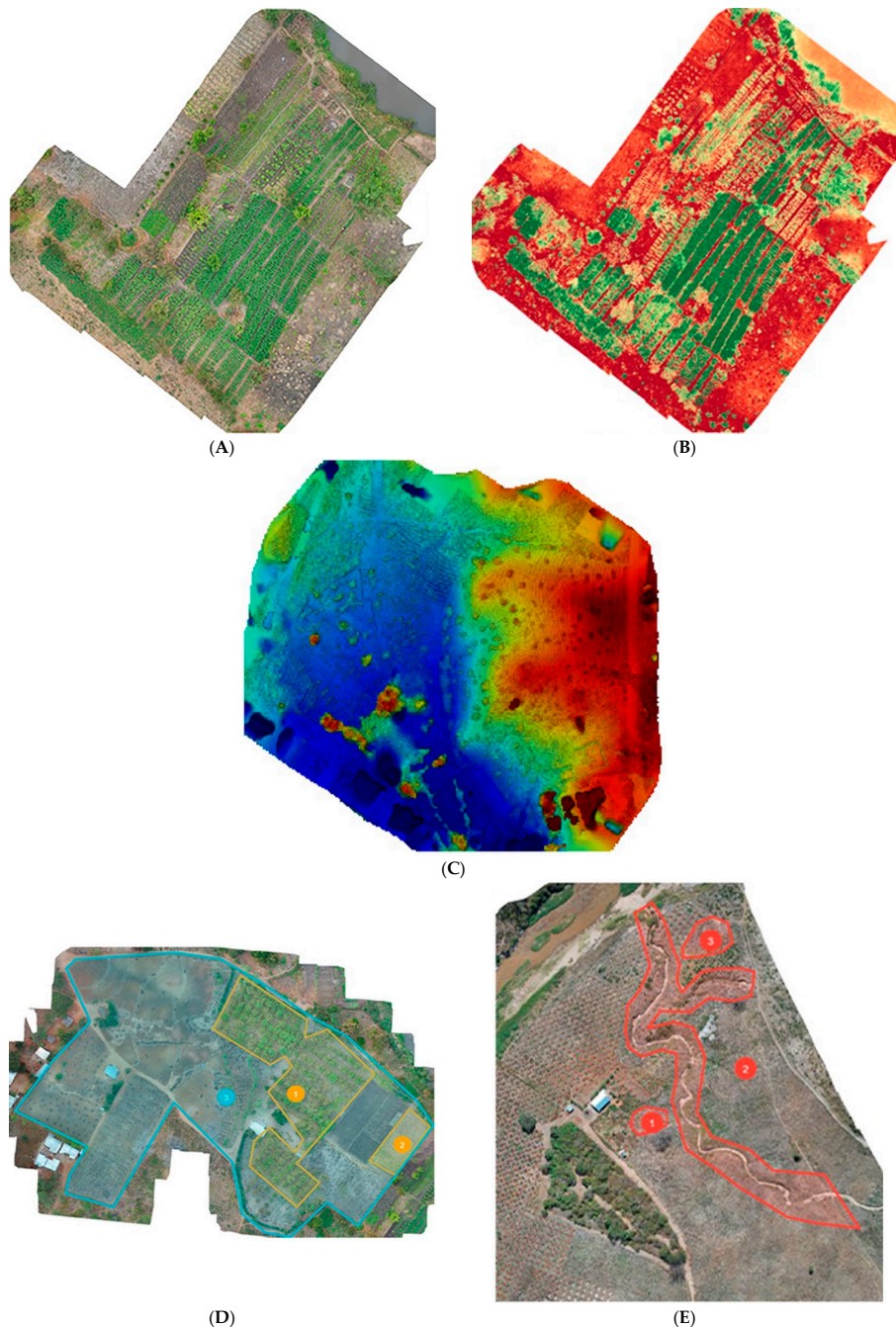

**Figure 2.** (**A**) Orthomosaic image and farm map, Neno district, Malawi (−15.4728191, 34.85927905); (**B**) plant health (VARI) farm profile, Neno district, Malawi (−15.4728191, 34.85927905). Reds indicate low vegetation growth, potentially due to stress or disease, while green indicates healthy, thriving vegetation with high chlorophyll content; (**C**) elevation profile of a farm, Neno district, Malawi (−15.4602466, 34.8420453). Reds indicate higher elevations and blues indicate lower elevations; (**D**) annotated land use image showing distribution of mangos, sugar cane, maize, and sweet potatoes, Neno district, Malawi (−15.4524496, 34.860456); (**E**) annotated soil erosion image, Neno district, Malawi (−15.5433168, 34.7350788).

**Table 2.** Distribution of on-farm issues identified in the drone images by the farmers.

| Issues Identified/Observed | Crop/Management Stage | Data Presented | Number of Farms | Proposed Solutions |
|---|---|---|---|---|
| Land management | - | Orthomosaic | 19 | Land-use planning |
| Nutrient deficiencies | Flowering | Plant health | 3 | Manure/Fertilizer application |
| Pest and diseases | Flowering/Fruiting | Plant health | 1 | Chemical application |
| Soil erosion and run-off | Land preparation | Orthomosaic and elevation | 16 | Construction of contour and box ridges, plant vetiver grass |
| Water logging | Flowering/Fruiting | Plant health and elevation | 5 | Improve soil structure by application of manure/fertilizer, construction of waterways, reduce watering frequency |
| Water stress | Flowering | Plant health and elevation | 5 | Increase watering frequency |

The results showed that the most observed issues identified by farmers from the drone images were related to land management, including crop management and rotation (19 farms), soil erosion/run-off (16 farms), water stress (5 farms), and nutrient deficiencies (3 farms). To validate the observations, ground verification was performed with an agricultural extension agent. Based on this information, farmers proposed potential solutions to address the identified issues. Solutions included the implementation of strategies such as the construction of contour and box ridges; planting of vetiver grass; and fertilizer application to mitigate soil erosion/run-off; the implementation of land-use planning for crop rotation and land management; and the adjustment of watering frequencies and application of manure/fertilizer to address water stress and nutrient deficiencies.

The data presented in this table underscores the value of conducting aerial farm surveys to identify previously undetected on-farm issues. The use of high-resolution images captured by drones enabled a more comprehensive analysis of the farms, and provided valuable insight to the farmers regarding the on-farm issues that would have otherwise remained unnoticed.

*3.3. Farmer Perceptions*

The study conducted a survey of farmers' perceptions and attitudes towards the use of drone technology in agriculture (Table 3). After interacting with drone imagery, a majority of farmers (70%) expressed interest in integrating drone technology into their agricultural practices. Notably, commercial farmers (100%) demonstrated greater interest than subsistence farmers (57%). Around 70% of the farmers were interested in utilizing drones for crop monitoring, with commercial farmers (100%) displaying a higher level of interest than subsistence farmers (57%). Additionally, more than 50% of the farmers expressed interest in employing drones for farm layout planning, with commercial farmers (100%) showing greater interest than subsistence farmers (29%).

The survey revealed that drone imagery had a significant impact on farm management decisions, with 100% of commercial farmers indicating that the technology would significantly impact their decision-making, compared to 71% of subsistence farmers. In terms of mitigating the challenges of climate change, 67% of commercial farmers believed that drone technology could help, compared to only 14% of subsistence farmers. Approximately 70% of all farmers believed that drone imagery would increase on-farm efficiencies and agriculture production, with commercial farmers (100%) showing higher interest than subsistence farmers (57%). The orthomosaic images and mapping were deemed the most helpful data type by the majority (87%) of farmers, with commercial farmers (100%) finding them more helpful than subsistence farmers (81%). Plant health data and elevation data were also found to be helpful by a majority of commercial farmers (77%) and subsistence farmers (60%), respectively.

**Table 3.** Farmers perceptions and attitudes toward drone technology.

| Perception/Attitude | Commercial | Subsistence | (n = 30) |
| --- | --- | --- | --- |
| Interest in incorporating drones in agriculture practice | 9 (100%) | 12 (57%) | 21 (70%) |
| Interest in using drones for crop monitoring | 9 (100%) | 12 (57%) | 21 (70%) |
| Interest in using drones for farm layout planning | 9 (100%) | 6 (29%) | 15 (50%) |
| Drone images would impact farm management decisions | 9 (100%) | 15 (71%) | 24 (80%) |
| Believe drone technology could help mitigate climate change | 6 (67%) | 3 (14%) | 9 (30%) |
| Believe drone imagery would increase on-farm efficiencies | 9 (100%) | 12 (57%) | 21 (70%) |
| Orthomosaic images and mapping helpful | 9 (100%) | 17 (81%) | 26 (87%) |
| Plant health data helpful | 9 (100%) | 14 (67%) | 23 (77%) |
| Elevation data helpful | 9 (100%) | 9 (43%) | 18 (60%) |
| Drone technology provides a new perspective | 9 (100%) | 17 (81%) | 26 (87%) |
| Drone technology an effective planning tool | 8 (88.9%) | 17 (81%) | 25 (83%) |
| Concerns about cost of drones | 9 (100%) | 21 (100%) | 30 (100%) |
| Concerns about marketing produce and managing profit margins | 9 (100%) | 0 (0%) | 9 (30%) |
| Do not trust the technology | 3 (33%) | 9 (43%) | 12 (40%) |
| Need more information | 8 (89%) | 14 (67%) | 22 (73%) |
| Believe using drones would help save on-farm/input costs | 6 (67%) | 3 (14%) | 9 (30%) |
| Open to using technology if implemented by agricultural extension agent | 9 (100%) | 18 (86%) | 27 (90%) |

The survey also revealed concerns about the cost of drones and the challenges associated with marketing produce and managing narrow profit margins when incorporating the technology into their on-farm operations. All farmers were concerned about the cost of drones. Only 30% of farmers believed that using drones would help save on-farm costs and agriculture inputs, with commercial farmers (100%) showing more belief in this statement than subsistence farmers (14%). Despite these concerns, the majority of the respondents (90%) indicated that they would be open to using the technology if implemented by a trained and experienced agricultural extension agent, underscoring the need for support and guidance from trained professionals.

*3.4. Data Sharing and Privacy*

A unique aspect of this study assessed smallholder farmers' views on data sharing and privacy (Table 4).

The results showed that 53% of the farmers were open to sharing their drone data with other farmers, extension workers, and researchers as they believed it could help improve agricultural practices and increase crop yields; however, only a small percentage of respondents felt comfortable sharing their data with government agencies and private companies, reflecting low levels of trust for these sectors. Moreover, only 10% of farmers felt adequately informed about data sharing laws and regulations, indicating the need for more education and awareness in this area. The study found that the most common source of information about drones was from friends, family, and/or acquaintances, with a majority of respondents stating they did not receive information from traditional sources such as print media, radio, and TV.

Thirty-five percent of farmers expressed concerns about privacy and the potential misuse of their data, which led to their hesitancy in sharing their data with outsiders. These farmers emphasized the importance of establishing clear guidelines for data sharing and ensuring that their data would be used only for the purposes they approved. Additionally, a majority of these farmers stated they would not feel comfortable using drone technology until policies and safeguards were in place.

Looking at the perception and attitude of farmers towards drone data sharing, commercial farmers (89%) were more comfortable sharing their data with government agencies compared to subsistence farmers (29%). Commercial farmers (100%) were more willing to share their data with other farmers, extension workers, and researchers than subsistence

farmers (33%). In terms of trust, commercial farmers (89%) were more trusting of government agencies to handle drone data responsibly compared to subsistence farmers (14%).

**Table 4.** Farmers perceptions and attitudes toward data sharing and privacy.

| Perception/Attitude | Commercial | Subsistence | (n = 30) |
|---|---|---|---|
| Comfortable sharing drone data with government agencies | 8 (29%) | 6 (29%) | 14 (47%) |
| Comfortable sharing drone data with private companies | 6 (67%) | 4 (19%) | 10 (33%) |
| Comfortable sharing data with other farmers, extension workers, researchers | 9 (100%) | 7 (33%) | 16 (53%) |
| Trust government agencies to handle drone data responsibly | 8 (89%) | 3 (14%) | 11 (37%) |
| Trust private companies to handle drone data responsibly | 6 (67%) | 2 (10%) | 8 (27%) |
| Understand the potential benefits of sharing drone data | 6 (67%) | 6 (29%) | 12 (40%) |
| Understand the potential risks of sharing drone data | 8 (89%) | 3 (14%) | 11 (37%) |
| Feel adequately informed about data sharing laws and regulations | 3 (33%) | 0 (0%) | 3 (10%) |
| Feel in control of how their drone data is being used | 4 (44%) | 3 (14%) | 7 (23%) |
| Feel their data privacy is adequately protected | 5 (56%) | 3 (14%) | 8 (27%) |
| Acquire knowledge about drones from friends, family, and/or acquaintances | 8 (89%) | 6 (29%) | 14 (47%) |
| Print media, radio, and/or TV is the main source of information about drones | 3 (33%) | 1 (5%) | 4 (13%) |
| Would share drone data if it meant lower input costs | 9 (100%) | 17 (81%) | 27 (90%) |
| Would share drone data if it meant improved crop yields | 9 (100%) | 19 (91%) | 28 (93%) |

The study also found that a significant number of commercial and subsistence farmers (100% and 89%, respectively) would be willing to share their drone data if it meant lower input costs. The majority of commercial farmers (100%) and subsistence farmers (91%) also stated they would share their data if it meant improved crop yields.

In conclusion, while many smallholder farmers are open to sharing their drone data, there is a need to address concerns about privacy and ensure that clear protocols are in place to protect personal information. Educating farmers on the benefits of data sharing and the measures in place to protect their privacy, as well as establishing clear guidelines for data use and sharing, could increase the uptake of drone technology among smallholder farmers and realize its full potential for improving agricultural practices and increasing crop yields.

## 4. Discussion

Drones have emerged as a key tool in the development of sustainable and precision agriculture systems and could have a far-reaching impact in sub-Saharan Africa where agriculture systems are inefficient and climate change and population growth pose a threat to food security [40,41]. However, the use of drones and drone imagery to inform on-farm decision making for smallholder farmers is still not well understood and technology uptake in the agriculture sector remains low. The results demonstrate the value and potential of drone data in providing detailed and accurate representations of smallholder farms, which can inform decisions about crop management, assess farming practices, and identify areas for improvement that can increase agricultural productivity. Moreover, the findings are consistent with other studies that have shown the potential of drones in improving agricultural productivity in other low-income countries, such as in Ghana, Nigeria, Uganda, and Namibia [42]. Similarly, the results, which demonstrate the value of drones in providing high-resolution maps and other data that can inform precision agriculture practices, are consistent with findings from other studies conducted in different regions of sub-Saharan Africa [21].

One of the main benefits of using drones is the high-definition maps that can be generated from drone imagery. Unlike satellite imagery, which often has a lower resolution and can be subject to cloud cover, drones allow for a more accurate and timely representation of smallholder farms. This is important because smallholder farms often have heterogeneous

landscapes, with a mix of crops, trees, and livestock, that traditional satellite imagery may not accurately represent. Maps and orthomosaics are particularly useful for precision agriculture, as it allows for a more comprehensive understanding of the health and distribution of crops and other vegetation [43]. In addition, orthomosaic images can be used to monitor changes in the landscape over time, providing valuable information for farm management decisions [44]. When interacting with orthomosaic images of their farms, the farmers in the study were able to assess the overall efficiency of their farming practices and identify areas for improvement in real time. Farmers stated that orthomosaic imagery help them make more informed decisions about crop management, and they stated they were likely to use this information to adjust their irrigation schedules, fertilization, and pest control, leading to an improvement in agricultural productivity and cost savings. In addition to the orthomosaic images, farmers expressed interest in plant health and elevation data, which can be easily generated from drone imagery using various software and can provide valuable information on crop performance, soil moisture, and potential crop yields. Farmers responded that this type of data could help them locate areas of their farm where yields are lower and allow them to proactively address potential problems. Farmers reported that elevation profiles aided their understanding of their farm's slope, drainage patterns, and potential water flow, which in turn can inform decisions about irrigation management, land preparation, and crop selection to optimize crop yields and minimize soil erosion [45].

The study's findings show that smallholder farmers in Malawi generally view drone usage and drone data positively when it comes to assisting with on-farm decision-making. They believe that this technology can help them make better-informed choices, boost efficiency and productivity, and save time and resources for specific tasks. However, the survey also uncovers some obstacles that impede the broad adoption of drones in Malawi's agricultural sector. Several farmers expressed concerns about the costs of the technology and the accuracy and interpretation of the data, and some farmers remained skeptical about the usefulness of the information provided by the drones, as well as the privacy and security of their personal information.

One of the main reasons for the skepticism is the limited education and low literacy rates among the farmers, a common limitation across sub-Saharan Africa [46], which can lead to confusion and mistrust about the technology and regulations. Many farmers surveyed expressed a lack of trust in the technology, and a significant number had difficulty comprehending applications of drones and drone data. The majority of farmers also indicated reluctance to share their data with government agencies or private companies and only a small percentage felt informed about drone regulations and data sharing laws. In order to support farmers with low literacy rates, it is crucial to provide them with clear and accessible information about drone technology and open data and data privacy regulations through educational materials, training programs, and community outreach initiatives. Radio, which is widely used for information dissemination across sub-Saharan Africa, could be a useful platform for reaching such farmers. Only a small percentage of farmers said they obtained information about drones via conventional media channels such as print media, radio, or TV. Those farmers who had knowledge of drones tend to have received information from friends, family, or acquaintances. To increase farmers' understanding and benefits of drone technology and data sharing and to increase transparency in the data collection and usage processes, it is important to educate them through collaboration with local organizations, such as non-profit groups, and agriculture extension agents. Trust is more likely to be established when information originates from reliable sources, including science organizations, educational institutions, and friends/family [47]. Local organizations can provide awareness, hands-on training, and dedicated support teams to help farmers understand the benefits and risks of drone technology and open data and data privacy.

In addition, the study findings reveal that both subsistence and commercial farmers show interest in incorporating drone technology into their farming practices, but with varying levels of interest in specific applications and concerns. This difference carries significant implications for policymakers and stakeholders in the agriculture sector. To

promote drone technology adoption in agriculture, policies and programs should consider these differences and develop strategies that cater to the unique needs and concerns of different types of farmers. For example, policies focusing on reducing drone costs or providing subsidies might be more effective in persuading commercial farmers to adopt the technology, while programs offering education and training on drone usage and benefits could be better suited for subsistence farmers. Policymakers and stakeholders must tailor their policies and programs to address the distinct needs and concerns of various types of farmers, considering their differing levels of interest, trust, and perceptions surrounding this technology.

While the results of this study suggest that the use of drones can improve decision-making for smallholder farmers in Malawi, it is important to note several limitations. The sample size was relatively small, and the study was conducted in a single district, which may limit the generalizability of the findings. Additionally, as an exploratory study, the focus was on gaining insights into the attitudes and perceptions of smallholder farmers towards drone technology in agriculture, and the study did not assess the actual impact of drone use on farm productivity or profitability. Furthermore, the decision not to use ground control points (GCPs) could limit the accuracy of the image processing and the resulting map. While statistical analysis could provide further insights into the findings, the primary goal was to generate qualitative data and explore the potential of drones in this context. Future studies could build on these insights and employ statistical analysis to further examine the relationships between variables. Further research is required to fully comprehend the long-term advantages of integrating drones into farm management planning and establish a suitable framework for their adoption. Finally, a primary concern is the cost of drone technology, particularly for smallholder farmers, and additional research is needed to determine how this cost can be addressed. However, by leveraging the expertise of agriculture extension workers, as has been demonstrated in other countries, this cost could be mitigated. With the appropriate support, smallholder farmers in sub-Saharan African countries, such as Malawi, can harness the potential benefits of drone technology and contribute to the development of sustainable and productive agriculture in the region.

## 5. Conclusions

This study offers a unique insight into Malawian smallholder farmers' attitudes towards drone technology and its potential benefits for decision-making in agriculture. It fills a knowledge gap in the adoption of emerging technologies in agriculture, particularly in sub-Saharan Africa. The findings have both theoretical and practical implications, informing policies and strategies to promote drone technology among smallholder farmers while addressing their needs and concerns. Ultimately, this could lead to increased agricultural productivity, cost savings, and improved food security in the region.

To harness the potential benefits of drone technology in sub-Saharan African countries like Malawi, it is crucial to leverage the expertise of agricultural extension workers, as demonstrated in other countries. Policymakers should consider investing in capacity-building programs for extension workers, providing subsidies for drone technology, and developing educational campaigns to raise awareness about the benefits and applications of drones in agriculture. By addressing the cost barrier and offering appropriate support, smallholder farmers can contribute to the development of sustainable and productive agriculture in the region, ultimately improving food security and livelihoods. This study highlights the importance of understanding farmers' perspectives and needs when designing and implementing policies and interventions to promote the adoption of emerging technologies such as drones in agriculture.

In addition to investing in capacity-building programs and providing subsidies for drone technology, policymakers should also prioritize the development of regulations that address data privacy and security concerns. By ensuring responsible use of drones in agriculture, farmers can feel confident in sharing their data and information, leading to more effective and efficient decision-making. It is essential to create policies that strike a

balance between protecting farmers' privacy while also promoting the use of open data to advance the agricultural sector. By taking these steps, policymakers can support the adoption of drone technology among smallholder farmers, ultimately leading to more sustainable and productive agriculture, improved food security, and enhanced economic prospects for communities in sub-Saharan Africa.

**Author Contributions:** Conceptualization, C.M., Y.N. and D.J.K.; methodology, C.M., Y.N., D.J.K., L.B.B., B.M. and C.C.; software, C.M., Y.N. and B.H.; validation, C.M., Y.N., D.J.K., L.B.B. and B.M.; formal analysis, C.M., Y.N. and D.J.K.; investigation, C.M., Y.N., D.J.K., L.B.B., B.M. and C.C.; resources, C.M., Y.N., D.J.K. and B.H.; data curation, C.M., Y.N. and B.H.; writing—original draft preparation, C.M., Y.N., D.J.K., L.B.B., B.M., C.C. and J.B.; writing—review and editing, C.M., Y.N., D.J.K., L.B.B., B.M., C.C. and J.B.; visualization, C.M., Y.N., D.J.K. and B.H.; supervision, C.M., Y.N., D.J.K., L.B.B., B.M., C.C., J.B. and B.H.; project administration, C.M., Y.N. and D.J.K.; funding acquisition, C.M. and B.H. All authors have read and agreed to the published version of the manuscript.

**Funding:** This work was supported by JSPS KAKENHI Grant Numbers (JP) 19H04362 (Risk assessment of the regional impact of the China "One-Belt-One-Road" (OBOR) project).

**Institutional Review Board Statement:** Not applicable.

**Data Availability Statement:** The data that support the findings of this study are available from the corresponding author upon reasonable request.

**Conflicts of Interest:** The authors declare no conflict of interest.

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
