# Peer review of "Can Drones Help Smallholder Farmers Improve Agriculture Efficiencies and Reduce Food Insecurity in Sub-Saharan Africa? Local Perceptions from Malawi"

_agriculture, doi:10.3390/agriculture13051075_

Round 1

Reviewer 1 Report

What type of analysis is expected by the drones by the participant is not clearly explained . 

could have used some statistical analysis for the results

can be accepted after slight modifications

Author Response

April 30, 2023

Thank you so much for your thoughtful and constructive feedback. We appreciate your time and believe our manuscript is much better as a result. Please see our point-by-point response to your comments below.

  1. We apologize for any confusion in the paper regarding the expected type of analysis to be conducted by the drones. In our study, the drones were primarily used to collect imagery of farmland, which was then analyzed using specialized software to provide data on various crop parameters such as plant health, growth patterns, elevation data. The collected data was then shared with smallholder farmers to inform farm management decisions and identify areas for potential improvement. We have added this information to the paper to provide greater clarity on the use of drones and the type of analysis expected from them. 
  2. As an exploratory study, the focus was on gaining insights into the attitudes and perceptions of smallholder farmers towards drone technology in agriculture. While statistical analysis could provide further insights into the findings, the primary goal was to generate qualitative data and explore the potential of drones in this context. Future studies could build on these insights and employ statistical analysis to further examine the relationships between variables. We have acknowledged the limitations of not conducting statistical analysis in the discussion section of the paper.

Based on your comments we have rewritten the discussion section as follows:

Lines 423-442:

While the results of this study suggest that the use of drones can improve decision-making for smallholder farmers in Malawi, it is important to note that the study has several limitations. For example, the sample size was relatively small, and the study was conducted in a single district, which may limit the generalizability of the findings. Additionally, as an exploratory study, the focus was on gaining insights into the attitudes and perceptions of smallholder farmers towards drone technology in agriculture and did not assess the actual impact of drone use on farm productivity or profitability. While statistical analysis could provide further insights into the findings, the primary goal was to generate qualitative data and explore the potential of drones in this context. Future studies could build on these insights and employ statistical analysis to further examine the relationships between variables. Further research is required to fully comprehend the long-term advantages of integrating drones into farm management planning and establish a suitable framework for their adoption.

Finally, a primary concern is the cost of drone technology, particularly for smallholder farmers, and additional research is needed to determine how this cost can be addressed. However, by leveraging the expertise of agriculture extension workers, as has been demonstrated in other countries, this cost could be mitigated. With the appropriate support, smallholder farmers in sub-Saharan African countries like Malawi can harness the potential benefits of drone technology and contribute to the development of sustainable and productive agriculture in the region.

Reviewer 2 Report

This paper shows a theme Drones for Help Smallholder Farmers in Malawi. The paper contributes to the advancement of knowledge. However, some points need to be better detailed for a complete understanding.

Title: it is very extensive and the study was regional. Adjust content and number of words.

Abstract: I suggest including analyzed year(s).

Keywords: include words other than the title.

Introduction: is comprehensive whit a good overview of problem in context.

Materials and Methods: Improve visual resolution and fit figure 1 on the page. Include the boundary of the Malawi. Comment on the smallholder total participating farmers (30) in relation to the total smallholder farmers in the region. Justify the sampling question and possible interpretations of the data. Indicate the possible sampling error. Include references/studies that support the proposed methodology. Especially for the interviews/questions and analysis. Include the questionnaire as a supplementary file to the paper.

Results: Include geographic coordinates of the areas in figure 2. Change to a single title indicating (A), (B), …. Adjust table 3. If possible, transform tables into graphs or include other analyzes of the data.

Discussions: Including more details with other aspects (authors/studies) in the discussions – compare the data obtained with other possible regions in Malawi or countries. For example, comment more on possible limitations of the drone in agriculture, use drones in smallholder farmers in different countries, ...

Conclusions: Detail more and include possible perspectives for future studies.

Author Response

Thank you for your thoughtful and constructive feedback. We appreciate your time and believe our manuscript is much better as a result. Please see our point-by-point response to your comments in the attachment. 

Reviewer 3 Report

this work explores sub-Saharan smallholder farmers perception about the use of  drones derived imagery to  support  and improve agriculture efficiencies. The manuscript is well structured and worth publishing. Nevertheless, some problems concerning drone derived imagery processing are present. In particular, is not clear how image blocks were oriented without using ground control points. Despite this was achieved in a dedicated software, parameters and algorithms adopted are crucial to critically assess and reproduce your Workflow. Therefore, I suggest a major revision of current manuscript.  

Author Response

April 30, 2023

Thank you for your thoughtful and constructive feedback. We appreciate your time and believe our manuscript is much better as a result. Please see our point-by-point response to your comments below. *Also attached as a word document.

  1. some problems concerning drone derived imagery processing are present. In particular, is not clear how image blocks were oriented without using ground control points. Despite this was achieved in a dedicated software, parameters and algorithms adopted are crucial to critically assess and reproduce your Workflow.

Response: Thank you for your comment. We acknowledge the importance of accurately orienting image blocks in drone-based surveys, and we apologize for not providing sufficient information on this aspect of our methodology. We understand that using ground control points (GCPs) can make the image processing and maps more accurate, but we decided not to use them because we didn't have enough time and resources. Instead, we made sure to fly the drone under the same weather conditions and at a steady height, speed, and overlap rate. Also, we used the DroneDeploy software, which has been proven to be reliable and accurate in other studies.

We have updated the methodology and discussion sections to highlight this limitation. We have also included a citation to support our approach.

Lines: 217-228

A DJI Mavic 2 Pro drone equipped with a high-resolution camera (20 MP with 1-inch CMOS) was employed for all flights, while flight planning and mapping were conducted using the DroneDeploy software (www.dronedeploy.com). The drone captured images of the farms from an elevation of 60 meters, with 70% front overlap and 65% side overlap at a drone speed of 6 m/s. The orientation of the image blocks was achieved using the DroneDeploy software without the use of ground control points. While we acknowledge that using ground control points (GCPs) can further enhance the accuracy of the image processing and the resulting map, we selected not to use GCPs due to the limited time and resources available. Nonetheless, we ensured that the drone flights were conducted under consistent weather conditions, and the drone was flown at a constant altitude, speed, and overlap rate. Additionally, the DroneDeploy software has been shown to produce accurate and reliable results in similar studies [35].

Lines: 490-497

While the results of this study suggest that the use of drones can improve deci-sion-making for smallholder farmers in Malawi, it is important to note several limitations. The sample size was relatively small, and the study was conducted in a single district, which may limit the generalizability of the findings. Additionally, as an exploratory study, the focus was on gaining insights into the attitudes and perceptions of smallholder farmers towards drone technology in agriculture, and the study did not assess the actual impact of drone use on farm productivity or profitability. Furthermore, the decision not to use ground control points (GCPs) could limit the accuracy of the image processing and the resulting map.

Reference

  1. Mulakala, J. Measurement Accuracy of the DJI Phantom 4 RTK & Photogrammetry. DroneDeploy, Published in Partnership with DJI 2019.

Reviewer 4 Report

This study investigated the role of drones in improving agricultural productivity by small farmers. The findings of this study showed that when smallholder farmers interact with drone data, they have a better understanding of their farm and are able to make more informed decisions that use fewer inputs and reduce production costs. Overall, this study addresses a topic of high relevance for research and also for practice. However, I believe some issues need revision and clarification.

Author Response

Thank you for your thoughtful and constructive feedback. We appreciate your time and believe our manuscript is much better as a result. Please see our point-by-point response in the attachment.

Round 2

Reviewer 3 Report

Authors have improved the manuscript according to my suggestions. Therefore I think that now the paper is ready to be published. 

Reviewer 4 Report

The authors have successfully addressed the comments. The revised manuscript can be accepted after a minor English language editing.

Minor editing of the English language required